**Data Availability Statement:** Personal health data underlying the findings of our study are not publicly available due to legal reasons related to

# Association between unemployment and the co-occurrence and clustering of common risky health behaviors: Findings from the Constances cohort

**Marie Plessz**[1,2]*, **Sehar Ezdi**[2], **Guillaume Airagnes**[3,4], **Isabelle Parizot**[2], **Céline Ribet**[4], **Marcel Goldberg**[4], **Marie Zins**[4], **Pierre Meneton**[5]

1 Centre Maurice Halbwachs (EHESS, ENS, CNRS, PSL), INRAE, Paris, France, 2 Centre Maurice Halbwachs (EHESS, ENS, CNRS, PSL), Paris, France, 3 Department of Psychiatry and Addictology, Hôpitaux Universitaires Paris Ouest, Paris, France, 4 UMS 011, Population-based Epidemiological Cohorts, Inserm, Villejuif, France, 5 UMR 1142 LIMICS, Inserm, Sorbonne Université, Université Paris 13, Paris, France

* Marie.plessz@inrae.fr

## Abstract

### Background

Unemployment is associated with a high prevalence of risky health behaviors. Mortality increases with the number of co-occurring risky behaviors but whether these behaviors co-occur with a greater than expected frequency (clustering) among unemployed people is not known.

### Methods

Differences according to unemployment status in co-occurrence and clustering of smoking, alcohol abuse, low leisure-time physical activity and unhealthy diet (marked by low fruit and vegetable intake) were assessed in 65,630 salaried workers, aged 18 to 65, who were participants in Constances, a French population-based cohort. Among them, 4573 (7.0%) were unemployed without (n = 3160, 4.8%) or with (n = 1413, 2.1%) past experience of unemployment.

### Results

Compared to the employed, unemployed participants without or with past experience of unemployment were similarly overexposed to each risky behavior (sex and age adjusted odds-ratios ranging from 1.38 to 2.19) except for low physical activity, resulting in higher rates of co-occurrence of two, three and four behaviors (relative risk ratios, RRR 1.20 to 3.74). Association between behavior co-occurrence and unemployment did not vary across gender, partnership status or income category. Risky behavior clustering, i.e., higher than expected co-occurrence rates based on the prevalence of each behavior, was similar across unemployment status. The same observations can be made in employed participants with past experience of unemployment, although overexposure to risky behaviors (ORs 1.15 to

data privacy protection. CONSTANCES has a data sharing policy but before data transfer a legal authorization has to be obtained from the CNIL (Commission nationale de l'informatique et des libertés), the French data privacy authority. The CONSTANCES email address is contact@constances.fr.

**Funding:** The French CONSTANCES Cohort is supported by the French National Research Agency (ANR-11-INBS-0002), Caisse Nationale d'Assurance Maladie des travailleurs salariés-CNAMTS and is funded by the Institut de Recherche en Santé Publique/Institut Thématique Santé Publique and the following sponsors: Ministère de la santé et des sports, Ministère délégué à la recherche, Institut national de la santé et de la recherche médicale, Institut national du cancer et Caisse nationale de solidarité pour l'autonomie. MP received funding from IReSP, general call for funding 2017 "prevention" (reference IReSP-17-PREV-25). The funders had no role in study design, data collection and analysis, decision to publish, or preparation of the manuscript.

**Competing interests:** The authors have declared that no competing interests exist.

**Abbreviations:** FV, fruit and vegetable.

1.38) and increased rates of co-occurrence (ORs 1.19 to 1.58) were not as pronounced as in the unemployed.

## Conclusions

Co-occurrence of risky behaviors in currently and/or formerly unemployed workers is not worsened by behavior clustering. Engagement in each of these behaviors should be considered an engagement in distinct social practices, with consequences for preventive policies.

## Introduction

Rising unemployment rates over the last decade in most European and North American countries [1] has attracted growing attention on the public health impact of job loss [2]. Indeed, unemployment is thought to raise premature mortality [3] by increasing the incidence of suicide [4], poor mental health [5], cancer [6, 7] and cardiovascular disease [8–12]. The mechanisms by which unemployment would increase the incidence of these pathologies remain elusive but overexposure to behavioral risk factors is likely to be involved [13, 14]. By far, the leading behavioral causes of premature mortality in Western populations are alcohol abuse, smoking, unbalanced diet and low physical activity [15].

Alcohol misuse was first suspected to be linked to unemployment during the industrial revolution in the 19th century [16]. Since then, the evidence suggesting that unemployed people are at increased risk of heavy alcohol intake, binge drinking and/or alcohol use disorders has accumulated. A review of the literature between 1990 and 2010 reports several studies showing that unemployment increases alcohol use and the incidence of alcohol disorders [16]. For example, in middle-aged Americans, becoming unemployed raises the risk of developing alcohol abuse/dependence six-fold compared to those who remain in employment [17]. More recently, positive and significant associations have been described between job loss during the past year and average daily ethanol consumption, number of binge drinking days and the probability of alcohol abuse and/or dependence diagnosis in large samples of the American population [18, 19]. In the Northern Swedish Cohort Study, non-moderate alcohol consumption in middle-aged adults has been associated with a higher exposure to unemployment during their youth [20]. Likewise, data from the Christchurch Health and Development Study in New Zealand showed that unemployment of at least three months' duration significantly increases the risk of alcohol use disorder in young adults [21].

A higher risk of smoking and/or increased frequency of tobacco use is another unhealthy behavior that has been convincingly linked to job loss and unemployment [16]. For example, unemployed middle-aged Americans consume more cigarettes per day if they already smoke and have a greater risk of relapse if they are ex-smokers in comparison to those who remain employed [22]. Unemployed young Americans who smoke are also less likely to attempt cessation than the employed [23]. More recent studies have confirmed that unemployment is associated with a higher risk of smoking in large samples of the American population [18, 24, 25]. This research has been corroborated by similar findings from the Scottish and German populations [26, 27].

Several studies suggest that unemployment may also lead to unhealthy food habits. For example, a long history of unemployment in Finnish young adults has been shown to be a good predictor of stress-related eating characterized by a high consumption of sausages, hamburgers, pizzas and chocolate [28]. Unemployment has been associated with low consumption

of starchy foods, fruits and vegetables, seafood and dairy products in a deprived middle-age French population attending food aid organizations [29]. Being unemployed has also been linked to food insecurity measured by the necessity of buying cheaper food and/or low consumption of fruits and vegetables in New Zealand [30]. Likewise, job insecurity/unemployment in Portuguese adults has been associated with an unhealthy dietary pattern characterized by low consumption of soups, vegetables, fresh fruits, fish, dairy products and high meat consumption [31]. Other studies have reported aggregated data showing that increased unemployment rates are associated with reduced consumption of fruits and vegetables and increased consumption of unhealthy foods such as snacks and fast food in North American populations [32, 33]. In Danish households, different dietary behaviors are observed depending on duration of unemployment, i.e., a higher intake of fat and protein due to increased consumption of animal-based foods immediately after job loss and a higher intake of carbohydrates and added sugar thereafter [34].

Low physical activity may be another unhealthy consequence of unemployment. Although physical activity at work varies substantially from one occupation to another [35], being unemployed has been associated with a reduction in daily physical activity in American adults [36] but not in Swedish adults [37]. Even leisure-time activity is modified among the unemployed. It has been shown that unemployed Swedish adults have lower leisure-time physical activity compared to the employed although the difference disappears when adjusting for education level [38]. Similarly, American adults who have been unemployed for a year or more [39], as well as young unemployed Americans [40], have less leisure-time physical activity than those who are employed. Compared to their employed counterparts, unemployed Finnish adults also more frequently report economic constraints and the lack of companionship as barriers for leisure-time physical activity [41].

The co-occurrence of behavioral risk factors dramatically increases the effects on health and mortality. Thus, old European men and women who combine adherence to a Mediterranean diet, moderate alcohol use, being physically active and non-smoking have a mortality rate one third of that which is observed in those who do not adopt any of these behaviors [42]. Similarly, middle-aged and older UK men and women who combine non-smoking, being physically active, moderate alcohol intake and fruit and vegetable intake of at least five servings a day have a 4-fold difference in the risk of dying over an average period of 11 years compared to those who do not adhere to any of these behaviors [43]. In addition to the mere co-occurrence of behavioral risk factors, there can exist a clustering of these factors, i.e., a higher frequency of co-occurrence than expected on the basis of the prevalence of each factor, that can exacerbate the effects on health and mortality [44]. To our knowledge, despite the large body of evidence showing that unemployment is associated with a high exposure to common behavioral risk factors, no study has examined the co-occurrence and clustering of these factors among unemployed people. One report has documented relative prevalence rates of cigarette smoking, risky drinking, non-engagement in leisure-time physical activity and low fruit/vegetable consumption among unemployed young US adults but has not explored the co-occurrence or clustering of these risky health behaviors [40]. This is the purpose of the present study, which investigates the co-occurrence and clustering of common unhealthy behaviors in French adults who are unemployed and/or have been unemployed in the past. The potential implication for health policies is to determine whether unemployed people need specific preventive strategies targeting reciprocal relationships between unhealthy behaviors, as would be the case if unemployment is associated with a clustering of these behaviors.

## Materials and methods

### Study population

The CONSTANCES cohort includes adults selected between 2012 and 2018 from the French population covered by the general health insurance system (over 85% of the population) according to a random sampling scheme stratified on age, sex, socioeconomic status and region [45]. The 15% excluded were mostly farmers or self-employed workers who had never worked as salaried workers. Inclusion criteria comprised of written informed consent, a comprehensive health examination in one of the 21 participating medical centers scattered across French metropolitan territory, and self-administered questionnaires on lifestyle, health-related behaviors, social and occupational conditions at inclusion and in the past (inclusion rate was 7.3%). The study received approval from both the Ethics Evaluation Committee of the French National Institute of Health and Medical Research (Inserm) and the National Committee for the Protection of Privacy and Civil Liberties (Cnil).

In the cohort at the time of data extraction, there were 91,259 adults aged 18 to 65, currently or previously salaried workers, who declared that they were currently either employed or unemployed and for whom lifestyle, occupational exposure and professional schedule questionnaires were completed. We excluded individuals who declared that they were not working for health reasons (149 unemployed) and who had missing data on covariates (12,172) or dependent variables (13,308). This resulted in a study population of 65,630 participants. Among them, 51,875 (79.1%) were never unemployed, 9,182 (14%) were unemployed in the past but not at inclusion, 3,160 (4.8%) were unemployed at inclusion but not in the past and 1,413 (2.1%) were unemployed both in the past and at inclusion.

### Exposure variable: Employment status

The employment status of participants at inclusion was assessed by a question with multiple choices, allowing participants to describe complex situations. Possible answers were: "I have a job (even if on sick leave, unpaid leave or availability, maternity, paternity, adoption or parental leave)", "Unemployed or job seeker", "Retired or no longer in business", "In training (pupil, student, trainee, apprentice, etc.)", "I do not work for health reasons (long-term illness, disability)", "No professional activity". Participants who ticked the box "I have a job", and only this one, were considered employed. Participants who ticked the box "Unemployed or job seeker" were considered unemployed only if they had not also ticked the boxes "I have a job" or "I do not work for health reasons". Lifetime unemployment was documented by a separate questionnaire in which participants were asked to report each time they had stopped working for a period of more than six months, and why (unemployment, health, other). By combining these data with those at inclusion, four types of experience of unemployment were defined: participants who were unemployed at inclusion and at least once in the past, those who reported being unemployed at least once in the past but not at inclusion, those who were unemployed at inclusion but reported no prior unemployment, and those who had a job and reported no earlier unemployment spell longer than 6 months. For the sake of clarity, we refer to the two last groups as "unemployed at inclusion only" and 'never unemployed' but it should be remembered that these participants might have had short periods of unemployment in the past.

### Outcome variables: Co-occurring behavioral risk factors

The main outcome variable is the number of behavioral risk factors each participant was exposed to. The four risk factors considered were collected in a self-completed questionnaire

at inclusion. Consumption of fruits and vegetables was determined through a food frequency questionnaire covering a regular week and used as a proxy for diet quality [46]. Data from this questionnaire have already been published [47]. People were considered at risk if the sum of their fruit (fresh or squeezed) and vegetable (raw or cooked) consumption was lower than three times per day, everyday (not just the day before the questionnaire). This cut-off is consistent with the definitions of low fruit and vegetable (FV) consumption in French health policy [48].

Leisure-time physical activity was determined by a calculated score ranging from 0 (i.e. being very active) to 6 (not being active at all). The physical activity questionnaire asked about regular practice of walking or cycling; practicing a sport; gardening or housekeeping over the past 12 months. Each of the three items was noted 0 if the answer was no; 1 if practice was regular but low (less than 15 minutes for sports, or 2 hours for the other two weekly); 2 if practice was regular and higher. Data regarding the score obtained by summing the three items has been published [49]. People with a score below three were considered at risk of low physical activity.

Smoking was understood as participants who smoked at inclusion, excluding consumption of e-cigarettes or cannabis. Alcohol abuse referred to drinking habits during the week before completing the questionnaire and was defined as a consumption exceeding two or three drinks per day in women and men respectively.

## Covariates

Age was divided in three categories (18–36, 37–47 and 48–65 years). Educational attainment was classified into three levels: university, secondary school, primary school. Income comprised monthly earnings of all household members and was collected in seven categories chosen according to the distribution of household disposable income in France in 2013 [50]. It was recoded as low (below 1500 euros), middle (between 1500 and 2800 euros) or high (above 2800 euros). A ´low´ household income assigned the participant to the first quintile of household income, while a 'high' household income assigned the participant above the median. Whether the household was single-headed and had children was also documented, as well as the region of residence grouped into six geographical areas (Paris area, north-east, south-west, south-east, Brittany, center area). As an indicator of overall physical and psychological health status of participants, self-rated health was assessed with an eight-level scale that was reduced to three levels for the analyses: good (levels 1 and 2), average (levels 3 and 4) or poor (levels 5 to 8 roughly corresponding to the 90th percentile).

## Statistical analyses

First, we computed descriptive statistics, including prevalence of each behavioral risk factor, by experience of unemployment.

Second, we analyzed co-occurrence. We defined co-occurrence as the number of risk factors the participants were exposed to. We estimated the association between risk co-occurrence at inclusion and experience of unemployment using multinomial logistic regressions. The base level of the outcome variable was "to be exposed to zero risk" and the reference category was "never unemployed". The models yielded relative risk ratios (RRR) for each number of co-occurring risk and each category of unemployment, as well as 95% confidence intervals and statistical tests for trends across unemployment category. Three models were applied: in M1 we minimally adjusted for sex and age; in M2 we added education, single-adult household, household with children, region of residence and self-rated health; in M3 we also added household monthly income category. We also tested interactions between experience of

unemployment and each covariate in order to check whether some groups of participants were more at risk of co-occurring behavioral risk factors.

Third, we examined risk clustering. Clustering of risk factors refers to the co-occurrences of risk factors with greater frequencies than expected by chance, i.e. if exposure to each risk were independent of one other [44]. For each number of co-occurring risk factors, expected rates of co-occurrence were computed from the prevalence of each risk factor assuming that they occurred independently. Clustering was defined as observed to expected prevalence ratios significantly >1. We also computed the frequency and clustering of each specific risk combinations. All the analyses were performed with the Stata software (version 15, Stata Corp., College Station, TX).

## Results

### Characteristics of participants at inclusion according to their experience of unemployment

As reported in Table 1, the characteristics of participants who were unemployed at inclusion without or with past experience of unemployment were very similar except for age, as those who were unemployed in the past were obviously more likely to be older. Compared to participants who never encountered unemployment, they were less educated and belonged more often to single-adult households without children and with low monthly income. Their geographical distribution was slightly different and they declared more often poor or average self-rated health.

Unemployed participants reported more often low FV intake. The minimally adjusted OR (95% CI) was 1.38 (1.25–1.53) for the unemployed at inclusion without past experience of unemployment. The unemployed were also more exposed to smoking (2.06 (1.91–2.23)) and alcohol abuse (1.78 (1.60–1.96)), but not low leisure-time physical activity (1.02 (0.94–1.10), p = 0.60, and 1.10 (0.97–1.23–1.31), p = 0.11). For the unemployed with past experience of unemployment, the OR were slightly but not significantly larger than for those without past experience of unemployment. Except for age, the characteristics of participants who were unemployed in the past but not at inclusion were generally intermediary, between those of participants who never encountered unemployment and those of participants who were unemployed at inclusion without or with past experience of unemployment. This also applied to low FV intake (1.14 (1.08–1.21)), smoking (1.38 (1.31–1.45)) and alcohol abuse (1.22 (1.13–1.32)). Exposure to low leisure-time physical activity was highest among those who were unemployed in the past but not at inclusion (1.20 (1.14–1.25)).

### Co-occurrence of risky health behaviors in participants at inclusion according to their experience of unemployment

Compared to participants who never encountered unemployment, those who were unemployed at inclusion without or with past experience of unemployment were similarly likely to be exposed to one risky behavior rather than none (minimally adjusted RRR 1.17 and 1.11) and more likely to be exposed to two (RRR 1.65 and 1.82), three (RRR 2.51 and 2.47) or four behaviors (RRR 2.95 and 3.74). This is shown in Table 2, model 1. There was a gradient in the association between exposure to unemployment and exposure to two, three or four risky behaviors, as shown by the highly significant p for trends.

As a result, the frequency ranking of co-occurrence of risky behaviors differed somewhat according to unemployment status (Table 3). For those exposed in the past and at inclusion, the most frequent was to be exposed to two behaviors (37.5%) followed by one (35.3%), three

**Table 1. Characteristics of participants at inclusion according to their experience of unemployment.**

| | | Experience of unemployment | | | | | | | | p |
|---|---|---|---|---|---|---|---|---|---|---|
| | | Never | | In the past only | | At inclusion only | | In the past & at inclusion | | |
| | | % | n | % | n | % | n | % | n | |
| - | All | 100.0 | 51,875 | 100.0 | 9182 | 100.0 | 3160 | 100.0 | 1413 | - |
| **Sex** | **Man** | 49.7 | 25,797 | 43.0 | 3944 | 48.2 | 1523 | 46.1 | 651 | <0.0001 |
| | **Woman** | 50.3 | 26,078 | 57.1 | 5238 | 51.8 | 1637 | 53.9 | 762 | |
| **Age (y)** | **18–36** | 32.6 | 16,907 | 22.7 | 2080 | 52.0 | 1644 | 31.3 | 442 | <0.0001 |
| | **37–47** | 33.9 | 17,579 | 35.6 | 3265 | 22.4 | 708 | 29.0 | 410 | |
| | **48–65** | 33.5 | 17,389 | 41.8 | 3837 | 25.6 | 808 | 39.7 | 561 | |
| **Education** | **Primary, lower secondary** | 15.6 | 8100 | 24.4 | 2237 | 26.7 | 845 | 26.6 | 376 | <0.0001 |
| | **High school diploma** | 14.3 | 7435 | 18.1 | 1660 | 19.7 | 623 | 22.4 | 316 | |
| | **University** | 70.1 | 36,340 | 57.6 | 5285 | 53.5 | 1692 | 51.0 | 721 | |
| **Single-adult household** | **No** | 77.6 | 40,242 | 73.5 | 6749 | 57.4 | 1814 | 60.2 | 850 | <0.0001 |
| | **Yes** | 22.4 | 11,633 | 26.5 | 2433 | 42.6 | 1346 | 39.8 | 563 | |
| **Household with children** | **No** | 41.4 | 21,496 | 42.3 | 3884 | 62.4 | 1972 | 56.1 | 793 | <0.0001 |
| | **Yes** | 58.6 | 30,379 | 57.7 | 5298 | 37.6 | 1188 | 43.9 | 620 | |
| **Household monthly income** | **Low** | 5.2 | 2678 | 10.4 | 958 | 38.8 | 1227 | 39.2 | 554 | <0.0001 |
| | **Middle** | 24.1 | 12,478 | 31.6 | 2897 | 32.4 | 1025 | 31.9 | 451 | |
| | **High** | 70.8 | 36,719 | 58.0 | 5327 | 28.7 | 908 | 28.9 | 408 | |
| **Region of residence** | **Paris area** | 17.9 | 9278 | 16.3 | 1492 | 22.5 | 711 | 20.2 | 285 | <0.0001 |
| | **North-east** | 16.0 | 8281 | 14.4 | 1321 | 14.4 | 456 | 15.2 | 214 | |
| | **South-west** | 19.7 | 10,208 | 21.3 | 1953 | 21.4 | 675 | 23.3 | 329 | |
| | **South-east** | 14.4 | 7487 | 15.8 | 1454 | 14.4 | 456 | 15.2 | 215 | |
| | **Brittany** | 15.5 | 8022 | 15.4 | 1412 | 13.7 | 432 | 12.2 | 173 | |
| | **Center area** | 16.6 | 8599 | 16.9 | 1550 | 13.6 | 430 | 13.9 | 197 | |
| **Self-rated health** | **Poor** | 6.8 | 3526 | 11.4 | 1048 | 14.5 | 457 | 16.3 | 230 | <0.0001 |
| | **Average** | 38.4 | 19,942 | 45.1 | 4140 | 43.5 | 1374 | 46.0 | 650 | |
| | **Good** | 54.8 | 28,407 | 43.5 | 3994 | 42.1 | 1329 | 37.7 | 533 | |
| **Low fruit & vegetable intake** | **No** | 20.2 | 10,473 | 19.9 | 1829 | 15.2 | 479 | 16.4 | 231 | <0.0001 |
| | **Yes** | 79.8 | 41,402 | 80.1 | 7353 | 84.8 | 2681 | 83.7 | 1182 | |
| **Smoking** | **No** | 79.7 | 41,334 | 75.4 | 6922 | 64.2 | 2030 | 65.4 | 924 | <0.0001 |
| | **Yes** | 20.3 | 10,541 | 24.6 | 2260 | 35.8 | 1130 | 34.6 | 489 | |
| **Low non-work physical activity** | **No** | 70.9 | 36791 | 67.4 | 6186 | 70.4 | 2224 | 69.1 | 977 | <0.0001 |
| | **Yes** | 29.1 | 15084 | 32.6 | 2996 | 29.6 | 936 | 30.9 | 436 | |
| **Alcohol abuse** | **No** | 91.0 | 47,188 | 89.6 | 8229 | 84.7 | 2676 | 84.4 | 1193 | <0.0001 |
| | **Yes** | 9.0 | 4687 | 10.4 | 953 | 15.3 | 484 | 15.6 | 220 | |

The percentages were calculated for each experience of unemployment and the differences between experiences were assessed by Chi-square test.

(14.4%) and zero (10%). For those never exposed, the most frequent was one behavior (46%), followed by two (30.5%) and zero (13.4%) with only 1.1% exposed to four behaviors.

When covariates were added to the minimally adjusted models, the RRR decreased but remained statistically significant for two, three or four risky behaviors. This was also true when income category was included. For example, for the unemployed with past experience of unemployment the RRR for two, three or four behaviors were 1.28, 1.39 and 1.78 respectively (Table 2, model 3).

**Table 2. Relative risk ratios of co-occurring risk factors according to experience of unemployment in Constances cohort: Multinomial regression (reference: 0 risk).**

| No. of risk factors | Experience of unemployment | M1 | | | M2 | | | M3 | | |
|---|---|---|---|---|---|---|---|---|---|---|
| | | RRR | 95% CI | p | RRR | 95% CI | p | RRR | 95% CI | p |
| 1 | Never | 1 | | | 1 | | | 1 | | |
| | Past only | 1.07 | 0.99–1.15 | 0.077 | 0.98 | 0.92–1.06 | 0.67 | 0.97 | 0.90–1.05 | 0.44 |
| | Now only | 1.17 | 1.03–1.34 | 0.018 | 1.07 | 0.94–1.22 | 0.33 | 1.02 | 0.88–1.16 | 0.83 |
| | Past & now | 1.11 | 0.92–1.34 | 0.280 | 1.00 | 0.83–1.22 | 0.98 | 0.94 | 0.78–1.15 | 0.56 |
| | *p for trend* | | | 0.172 | | | 0.76 | | | 0.68 |
| 2 | Never | 1 | | | 1 | | | 1 | | |
| | Past only | 1.3 | 1.21–1.40 | <0.0001 | 1.12 | 1.04–1.21 | 0.003 | 1.09 | 1.01–1.18 | 0.023 |
| | Now only | 1.65 | 1.45–1.89 | <0.0001 | 1.34 | 1.17–1.54 | 0.00002 | 1.20 | 1.04–1.38 | 0.011 |
| | Past & now | 1.82 | 1.50–2.20 | <0.0001 | 1.46 | 1.20–1.77 | 0.0001 | 1.28 | 1.05–1.56 | 0.013 |
| | *p for trend* | | | <0.0001 | | | <0.0001 | | | 0.007 |
| 3 | Never | 1 | | | 1 | | | 1 | | |
| | Past only | 1.62 | 1.48–1.78 | <0.0001 | 1.29 | 1.17–1.42 | <0.0001 | 1.24 | 1.13–1.36 | <0.0001 |
| | Now only | 2.51 | 2.16–2.93 | <0.0001 | 1.67 | 1.43–1.95 | <0.0001 | 1.44 | 1.23–1.69 | <0.0001 |
| | Past & now | 2.47 | 1.98–3.08 | <0.0001 | 1.65 | 1.31–2.07 | <0.0001 | 1.39 | 1.11–1.75 | 0.005 |
| | *p for trend* | | | <0.0001 | | | <0.0001 | | | 0.002 |
| 4 | Never | 1 | | | 1 | | | 1 | | |
| | Past only | 1.86 | 1.53–2.25 | <0.0001 | 1.40 | 1.15–1.69 | 0.001 | 1.32 | 1.09–1.61 | 0.005 |
| | Now only | 2.95 | 2.26–3.86 | <0.0001 | 1.71 | 1.30–2.25 | 0.0001 | 1.42 | 1.07–1.89 | 0.016 |
| | Past & now | 3.74 | 2.59–5.40 | <0.0001 | 2.19 | 1.51–3.19 | <0.0001 | 1.78 | 1.21–2.62 | 0.003 |
| | *p for trend* | | | <0.0001 | | | <0.0001 | | | 0.004 |

Multinomial regression. RRR: relative risk ratio. The base level of the outcome is: exposed to zero risk.

M1: adjusted on sex and age

M2: M1 + self-rated health, education, partnership status, presence of children, region

M3: M2 + income category

When we examined interactions between covariates and experience of unemployment in the fully adjusted model, there were statistically significant interactions for age (chi-square test p = 0.015), self-rated health (p = 0.052) and education (p = 0.036). However, when we stratified models over these variables, there was no pattern or trend in the RRR (S2 Table). Association between risky behavior co-occurrence and unemployment did not vary across gender, partnership status, or income category.

## Risky health behavior clustering according to experience of unemployment

Table 3 shows evidence of behavior clustering, whatever the experience of unemployment: the observed numbers of participants with none, three or four risky behaviors was greater than expected. The numbers of participants with one or two behaviors were smaller.

The lack of participants having one or two risky behaviors was similar in all experiences of unemployment. The excess of participants having no risky behavior (as compared to expected numbers) was higher when they were unemployed at inclusion without or with past experience of unemployment than when they never encountered unemployment. Conversely, the greater-than-expected exposure to three or four risky behaviors tended to be lower for participants having experienced unemployment of any kind, suggesting that having experienced unemployment was associated with smaller behavior clustering. However, the differences between O/E ratios across experience of unemployment only reached statistical significance for the

**Table 3. Clustering of behavioral risk factors in participants at inclusion according to their experience of unemployment (observed-to-expected ratios).**

| No. of occurring risk factors | Experience of unemployment | Observed | | Expected | | O/E (95% CI) |
|---|---|---|---|---|---|---|
| | | % | n | % | n | |
| 0 | Never | 13.4 | 6971 | 10.4 | 5382 | 1.29 (1.26–1.33) |
| | In the past only | 12.6 | 1158 | 9.1 | 833 | 1.39 (1.31–1.47) |
| | At inclusion only | 9.1 | 288 | 5.8 | 183 | 1.57 (1.40–1.77) |
| | In the past & at inclusion | 10.0 | 141 | 6.2 | 88 | 1.60 (1.35–1.89) |
| 1 | Never | 46.0 | 23,882 | 48.9 | 25,392 | 0.94 (0.93–0.95) |
| | In the past only | 41.7 | 3831 | 44.9 | 4120 | 0.93 (0.90–0.96) |
| | At inclusion only | 37.1 | 1171 | 39.2 | 1239 | 0.94 (0.89–1.00) |
| | In the past & at inclusion | 35.3 | 499 | 39.2 | 553 | 0.90 (0.82–0.98) |
| 2 | Never | 30.5 | 15,830 | 33.1 | 17,191 | 0.92 (0.91–0.93) |
| | In the past only | 32.7 | 2998 | 36.1 | 3310 | 0.91 (0.87–0.94) |
| | At inclusion only | 35.4 | 1120 | 40.0 | 1264 | 0.89 (0.83–0.94) |
| | In the past & at inclusion | 37.5 | 530 | 39.6 | 560 | 0.95 (0.87–1.03) |
| 3 | Never | 8.9 | 4596 | 7.1 | 3689 | 1.25 (1.21–1.28) |
| | In the past only | 11.4 | 1045 | 9.3 | 858 | 1.22 (1.14–1.29) |
| | At inclusion only | 15.9 | 504 | 13.6 | 430 | 1.17 (1.07–1.28) |
| | In the past & at inclusion | 14.4 | 204 | 13.6 | 192 | 1.06 (0.92–1.22) |
| 4 | Never | 1.1 | 596 | 0.4 | 221 | 2.70 (2.48–2.92) |
| | In the past only | 1.6 | 150 | 0.7 | 61 | 2.46 (2.08–2.88) |
| | At inclusion only | 2.4 | 77 | 1.4 | 43 | 1.79 (1.41–2.24) |
| | In the past & at inclusion | 2.8 | 39 | 1.4 | 20 | 1.95 (1.39–2.67) |

Observed and expected prevalence rates of single or co-occurring risk factors, which included low fruit and vegetable intake, smoking, low leisure-time physical activity and alcohol abuse, are reported in the table with observed to expected ratios (O/E) and 95% confidence intervals (95% CI) calculated on the basis of the frequencies (n) for each experience of unemployment.

exposure to four risky behaviors. In summary, risky behavior clustering was observed across all experiences of unemployment but was not stronger among those with past and/or present exposure of unemployment than in those who never experienced unemployment.

As shown in Tables 3 and 4, the three most frequent combinations of risky behaviors were low FV intake (30.9 to 40.3%), low FV intake and low physical activity (13.7 to 18.7%), and low FV intake and smoking (9.3 to 16.1%), with slight variations in the frequency order across experience of unemployment. The two combinations with the highest observed-to-expected ratios were all four behaviors (O/E 1.95 to 2.70), and FV & smoking & alcohol abuse (1.63 to 1.92). Beyond that, the order of risky behavior combinations varied across experience of unemployment.

## Discussion

The present study reports the occurrence, co-occurrence and clustering of common risky behaviors in adults who were unemployed at the time of the analyses and/or had encountered unemployment in the past.

Prevalence rates of exposure to none or only one risky behavior were at the higher end of the distributions described in other Western populations while rates of exposure to two, three or four behaviors were at the lower end [51–59]. The most frequent co-occurrences of two risky behaviors were low FV intake associated with either smoking, leisure-time physical inactivity or alcohol abuse, in agreement with what has been reported in the English population

**Table 4. Combinations of behavioral risk factors in participants at inclusion according to their experience of unemployment.**

| Risk factors | | Experience of unemployment | | | | | | | |
|---|---|---|---|---|---|---|---|---|---|
| | | Never | | In the past only | | At inclusion only | | In the past & at inclusion | |
| | | % | n | % | n | % | n | % | N |
| Fv | O | 40.3 | 20,922 | 35.6 | 3265 | 32.3 | 1020 | 30.9 | 436 |
| | E | 41.0 | 21,276 | 36.5 | 3348 | 32.5 | 1027 | 31.9 | 451 |
| | O/E (95% CI) | 0.98 (0.97–1.00) | | 0.97 (0.94–1.01) | | 0.99 (0.93–1.06) | | 0.97 (0.87–1.06) | |
| Sm | O | 1.7 | 886 | 2.1 | 195 | 2.1 | 65 | 1.6 | 23 |
| | E | 2.6 | 1373 | 3.0 | 272 | 3.2 | 102 | 3.3 | 47 |
| | O/E (95% CI) | 0.64 (0.60–0.69) | | 0.72 (0.62–0.82) | | 0.64 (0.49–0.81) | | 0.49 (0.31–0.73) | |
| Pi | O | 3.3 | 1698 | 3.3 | 301 | 2.1 | 66 | 2.3 | 32 |
| | E | 4.3 | 2209 | 4.4 | 403 | 2.4 | 77 | 2.8 | 39 |
| | O/E (95% CI) | 0.77 (0.73–0.81) | | 0.75 (0.66–0.84) | | 0.86 (0.66–1.09) | | 0.82 (0.56–1.16) | |
| Al | O | 0.7 | 376 | 0.8 | 70 | 0.6 | 20 | 0.6 | 8 |
| | E | 1.0 | 535 | 1.1 | 96 | 1.0 | 33 | 1.2 | 16 |
| | O/E (95% CI) | 0.70 (0.63–0.78) | | 0.73 (0.57–0.92) | | 0.61 (0.37–0.94) | | 0.50 (0.22–0.98) | |
| Fv & Sm | O | 9.3 | 4803 | 10.2 | 937 | 16.1 | 508 | 15.9 | 225 |
| | E | 10.5 | 5426 | 11.9 | 1093 | 18.1 | 572 | 16.9 | 239 |
| | O/E (95% CI) | 0.88 (0.86–0.91) | | 0.86 (0.80–0.91) | | 0.89 (0.81–0.97) | | 0.94 (0.82–1.07) | |
| Fv & Pi | O | 17.1 | 8874 | 18.3 | 1678 | 13.7 | 433 | 16.0 | 226 |
| | E | 16.8 | 8732 | 17.6 | 1620 | 13.7 | 432 | 14.3 | 202 |
| | O/E (95% CI) | 1.02 (0.99–1.04) | | 1.04 (0.99–1.09) | | 1.00 (0.91–1.10) | | 1.12 (0.98–1.27) | |
| Fv & Al | O | 3.2 | 1648 | 3.2 | 290 | 4.5 | 141 | 3.9 | 55 |
| | E | 4.1 | 2113 | 4.2 | 388 | 5.9 | 186 | 5.9 | 83 |
| | O/E (95% CI) | 0.78 (0.74–0.82) | | 0.75 (0.66–0.84) | | 0.76 (0.64–0.89) | | 0.66 (0.50–0.86) | |
| Sm & Al | O | 0.3 | 153 | 0.3 | 27 | 0.4 | 14 | 0.6 | 8 |
| | E | 0.3 | 136 | 0.3 | 31 | 0.6 | 18 | 0.6 | 9 |
| | O/E (95% CI) | 1.12 (0.95–1.32) | | 0.87 (0.57–1.27) | | 0.78 (0.42–1.30) | | 0.89 (0.38–1.75) | |
| Sm & Pi | O | 0.5 | 263 | 0.5 | 50 | 0.5 | 17 | 0.9 | 13 |
| | E | 1.1 | 563 | 1.4 | 132 | 1.4 | 43 | 1.5 | 21 |
| | O/E (95% CI) | 0.47 (0.41–0.53) | | 0.38 (0.28–0.50) | | 0.39 (0.23–0.63) | | 0.62 (0.33–1.06) | |
| Pi & Al | O | 0.2 | 89 | 0.2 | 16 | 0.2 | 7 | 0.2 | 3 |
| | E | 0.4 | 219 | 0.5 | 47 | 0.4 | 14 | 0.5 | 7 |
| | O/E (95% CI) | 0.41 (0.33–0.50) | | 0.34 (0.19–0.55) | | 0.50 (0.20–1.03) | | 0.43 (0.09–1.25) | |
| Fv & Sm & Al | O | 2.0 | 1032 | 2.7 | 244 | 5.3 | 168 | 5.7 | 81 |
| | E | 1.0 | 539 | 1.4 | 127 | 3.3 | 103 | 3.1 | 44 |
| | O/E (95% CI) | 1.91 (1.80–2.03) | | 1.92 (1.69–2.18) | | 1.63 (1.39–1.90) | | 1.84 (1.46–2.29) | |
| Fv & Sm & Pi | O | 5.3 | 2771 | 7.0 | 645 | 8.8 | 279 | 6.9 | 97 |
| | E | 4.3 | 2227 | 5.8 | 529 | 7.6 | 240 | 7.6 | 107 |
| | O/E (95% CI) | 1.24 (1.20–1.29) | | 1.22 (1.13–1.32) | | 1.16 (1.03–1.31) | | 0.91 (0.73–1.11) | |
| Fv & Pi & Al | O | 1.5 | 756 | 1.6 | 144 | 1.7 | 55 | 1.6 | 23 |
| | E | 1.7 | 867 | 2.0 | 188 | 2.5 | 78 | 2.6 | 37 |
| | O/E (95% CI) | 0.87 (0.81–0.94) | | 0.77 (0.65–0.90) | | 0.70 (0.53–0.92) | | 0.62 (0.39–0.93) | |
| Sm & Pi & Al | O | 0.1 | 37 | 0.1 | 12 | 0.1 | 2 | 0.2 | 3 |
| | E | 0.1 | 56 | 0.2 | 15 | 0.2 | 8 | 0.3 | 4 |
| | O/E (95% CI) | 0.66 (0.46–0.91) | | 0.80 (0.41–1.40) | | 0.25 (0.03–0.90) | | 0.75 (0.15–2.19) | |

Observed (O) and expected (E) prevalence rates of combinations of risk factors, which included low fruit and vegetable intake (Fv), smoking (Sm), low leisure-time physical activity (Pi) and alcohol abuse (Al), are reported in the table with observed to expected ratios (O/E) and 95% confidence intervals (95% CI) calculated on the basis of the frequencies (n) for each experience of unemployment.

[60]. The most common exposures to three risky behaviors, i.e., low FV intake and smoking associated with either leisure-time physical inactivity or alcohol abuse were also among those that have been the most frequently observed in other European populations [54, 57, 59]. Likewise, greater than expected proportions of people exposed to none, two, three or four risky behaviors have already been documented in several Western populations [53, 54, 57, 59, 61, 62].

Unemployment at inclusion (~7% rate, slightly lower than the current national average which is around 9% but very close to the mean European rate) was correlated with significant differences in the frequency of co-occurrence of risky behaviors. The first observation is that unemployment was associated with higher prevalence rates of each behavior, adding to the large body of evidence already discussed in the introduction. The difference was quite substantial for smoking and alcohol abuse (1.7-fold in both cases), in agreement with the literature which has reported average increases of 1.6 and 2.2-fold respectively in unemployed people [16]. In contrast, the difference was much more limited for low FV intake and low leisure-time physical activity (less than 10% in both cases). Low FV intake was very frequent, which could explain this small increase. It is more difficult to account for the small variation in leisure-time physical activity whose prevalence has been shown in some studies to be 1.6-fold higher in unemployed people [38, 39].

As a consequence of the higher prevalence of each risky behavior, unemployed participants were more frequently exposed to two, three or four behaviors, resulting in a further increase in their mortality risk [43]. This was true even after controlling for income category, indicating that income differences contribute partly but not entirely to the association between unemployment and risky behaviors. Stratified analyses indicated that among unemployed participants, there was no variation in behavior co-occurrence across gender, household status or income category. Clustering of risky behaviors was observed, in the sense that the observed prevalence of zero, three or four behaviors were higher than those expected if exposures had been statistically independent. However, clustering was similar across experience of unemployment, or even slightly weaker among the unemployed.

Essentially, similar results were obtained in unemployed participants without or with past experience of unemployment, suggesting that potential effects of current unemployment on the engagement in risky behaviors prevail over those of past unemployment. However, most of the observations can also be made in employed participants who have been unemployed in the past, although to a smaller extent than in unemployed participants. This supports the view that the increased prevalence and co-occurrence of risky behaviors would be a consequence of unemployment and that this might contribute to poor health in unemployed people [2, 3], although analyses on prospective data are required to support this assumption.

The present study has several limitations. First, the study selected only salaried or formerly salaried participants, excluding self-employed or farmers (who were not covered by the French unemployment insurance system at the time of the study). Second, the very low participation rate resulted in the selection of motivated and socially privileged individuals even though the stratified sampling strategy tried to compensate for the higher non-response of people with a low socioeconomic status. This is illustrated by a comparison of cohort participants with randomly selected workers in the same age range [63]. The proportion of participants with university education is very high and the unemployment rate is somewhat lower (S1 Table). Third, unemployment status and risky behaviors were self-reported. Reporting bias is possible and may vary across population categories. More specifically some experiences of unemployment were not captured by the questionnaire, such as unemployment upon labor-market entry (the questionnaire starts with the first job lasting at least 6 months) or short alternating episodes of unemployment and paid jobs (the questionnaire records episodes lasting at least six months).

The number and actual duration of episodes of unemployment in the past or at inclusion were also not available for the analyses. Last, the cutoff points for exposure to risky behaviors, although chosen consistently with French public health guidelines, might affect the results.

In conclusion, this study shows that current unemployment, and to a lesser extent past unemployment, was associated with increased engagement in common risky health behaviors, supporting the view that these behaviors might partly mediate increased morbidity and mortality in unemployed people [3]. However, we found that risky behaviors did not cluster more among participants experiencing unemployment than among those in employment. This might be due to the fact that engagement in risky behaviors relies on mechanisms that differ from one behavior to another, hence, the prevalence of one behavior can increase independently of others, regardless of the employment status. Engagement in risky behaviors may be related to strategies for coping with stress but may also involve multiple understandings of when and how much it is appropriate to consume alcoholic drinks, cigarettes or healthy foods, or engage in any practice that generates physical activity [64]. This suggests that preventive strategies should be adapted to each unhealthy behavior among unemployed as well as employed people, and would benefit from a practice-based approach, thereby acknowledging that they constitute distinct social practices with specific meaning, context, and embeddedness in social relations [65].

## Supporting information

**S1 Table. Baseline sociodemographic characteristics of CONSTANCES participants compared to randomly selected workers.**
(DOCX)

**S2 Table. Relative risk ratios of co-occurring risk factors according to experience of unemployment in Constances cohort: Stratification by age, education and self-rated health, multinomial regression (reference: 0 risk).**
(DOCX)

## Author Contributions

**Conceptualization:** Marie Plessz, Guillaume Airagnes, Isabelle Parizot, Pierre Meneton.

**Data curation:** Marie Plessz.

**Formal analysis:** Marie Plessz, Pierre Meneton.

**Methodology:** Marie Plessz, Marie Zins, Pierre Meneton.

**Project administration:** Marie Plessz, Céline Ribet.

**Resources:** Céline Ribet, Marcel Goldberg, Marie Zins.

**Writing – original draft:** Marie Plessz, Pierre Meneton.

**Writing – review & editing:** Marie Plessz, Sehar Ezdi, Guillaume Airagnes, Isabelle Parizot, Céline Ribet, Marcel Goldberg, Marie Zins, Pierre Meneton.

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
