## [Decision Letter · Decision Letter 0]

20 Nov 2019

PONE-D-19-26293

Association between unemployment and the co-occurrence and clustering of common risky health behaviors: Findings from the Constances cohort

PLOS ONE

Dear Dr Plessz,

Thank you for submitting your manuscript to PLOS ONE. After careful consideration, we feel that it has merit but does not fully meet PLOS ONE’s publication criteria as it currently stands. Therefore, we invite you to submit a revised version of the manuscript that addresses the points raised during the review process.

We would appreciate receiving your revised manuscript by Jan 04 2020 11:59PM. To enhance the reproducibility of your results, we recommend that if applicable you deposit your laboratory protocols in protocols.io, where a protocol can be assigned its own identifier (DOI) such that it can be cited independently in the future. For instructions see: http://journals.plos.org/plosone/s/submission-guidelines#loc-laboratory-protocols

A **rebuttal letter** that responds to **EACH** point raised by the academic editor and reviewer(s). This letter should be uploaded as separate file and labeled 'Response to Reviewers'.A **marked-up copy** of your manuscript that highlights changes made to the original version. This file should be uploaded as separate file and labeled 'Revised Manuscript with Track Changes'.An **unmarked version** of your revised paper without tracked changes. This file should be uploaded as separate file and labeled 'Manuscript'.

We look forward to receiving your revised manuscript.

Kind regards,

Brecht Devleesschauwer

Academic Editor

PLOS ONE

Journal Requirements:

Additional Editor Comments:

The reviewers expressed several major concerns about the current manuscript. I would like to give the authors the chance to respond to these concerns and make relevant chances. However, please note that at this stage we cannot guarantee that your revised manuscript will be accepted for publication.

In your revision note, please include EACH comment of the editor/reviewers, provide your reply, and when relevant, include the modified/new text (or motivate why you decided not to modify the text).

Reviewer #1 stressed the fact that the results section is particularly long and very difficult to follow. The use of multinomial regression instead of logistic regression should be considered. The authors were also asked to elaborate more on potential mechanisms, as this is not emphasized sufficiently in the discussion.

Reviewer #4 had made a comment about the novelty of the results. Please note that the PLOS ONE criteria for publication do not include an assessment of scientific novelty or innovation (https://journals.plos.org/plosone/s/criteria-for-publication). This comment will therefore not affect our final conclusion.

Reviewers' comments:

Reviewer's Responses to Questions

**Comments to the Author**

1. Is the manuscript technically sound, and do the data support the conclusions?

Reviewer #1: Partly

Reviewer #2: Partly

Reviewer #3: Yes

Reviewer #4: Yes

2. Has the statistical analysis been performed appropriately and rigorously? 

Reviewer #1: I Don't Know

Reviewer #2: Yes

Reviewer #3: I Don't Know

Reviewer #4: No

3. Have the authors made all data underlying the findings in their manuscript fully available?

Reviewer #1: Yes

Reviewer #2: Yes

Reviewer #3: Yes

Reviewer #4: Yes

4. Is the manuscript presented in an intelligible fashion and written in standard English?

Reviewer #1: Yes

Reviewer #2: Yes

Reviewer #3: Yes

Reviewer #4: No

5. Review Comments to the Author

Reviewer #1: In this manuscript, results are reported of a cross-sectional study evaluating the association between unemployment and the co-occurrence and clustering of common risky health behaviors. The authors used data from more than 65.000 men and women from the French Constances cohort. They found that current unemployment, and to a lesser extent past unemployment, was associated with increased engagement in common risky behaviors. However, unhealthy behaviors did not cluster among men or women experiencing unemployment. Apart from the results, the manuscript is well written. Most of the results, however, are presented less clearly and succinctly. Below are a few comments and questions for the authors’ consideration.

- The sample is restricted to complete data. The authors should provide information how many individuals are dropped because of missing data. The authors should also discuss potential problems due to missing at random (MAR) and not missing at random (NMAR). What is also unclear is that in the discussion it is stated that self-employed individuals and inactive populations (I would not define students and retirees as inactive) were excluded, but no mention of this was made in the methods section and it is unclear why these people were excluded. Furthermore, what was done with disabled people or individuals who had no income?

- Confounding variables are controlled for in the cross-sectional analyses. The authors should discuss why they only applied one model, rather than two or three. For instance, income is included in the model, but income is likely to be one of the mechanisms how unemployment affects health behaviour. To study whether this is the cause, it is plausible to include this variable in an additional model and then discuss whether results do or do not change when including this variable. Furthermore, it should be discussed that important confounding variables (e.g. education and BMI) are missing.

- From the methods section it remains unclear what the exposure and outcome is. Please clarify and it would help as the exposure and outcome are described in a consistent order.

- Separate analyses were done for men and women, but no tests for interaction were made and reported. I would suggest to test this first, before reporting the results separately. Furthermore, I would recommend testing for interaction with income and age too (and eventually reporting the separated analyses if necessary).

- The current health behavior variables are very crude. What is the reason for that no distinctions are made with non-smokers or ex-smokers (in the case of smoking) or with low, moderate or high physical activity (in the case of physical activity) and with low risk drinker, non-drinker, rarely drinker and risky drinker (in the case of alcohol consumption) and other variables? I recommend the authors to include more detailed exposure variables.

- It is unclear to me why the authors used logistic regression analyses. Why not using a multinomial logistic regression? The multinomial model is an extension of the logistic regression model that allows for more than two categories in the response variable (i.e., number of risky health behaviours). Furthermore, what is the P for trend?

- The results section is very long and difficult to follow. Some sentences include more than 8 sentences and too many numbers and ORs are presented. Please restructure and only present the main findings. Furthermore, the odds ratios presented on page 10 are not included in a table. From table 2 it appears that the O/E of risky health behavior is higher in those who have never been unemployed. I find that hard to understand, shouldn’t that be the other way around? Please explain.

- The mechanisms explained in the discussion are too general. Specific arguments are missing: arguments should be provided why unemployment is associated with smoking and unhealthy diet as the associations are not obvious, e.g. as unemployment may also provide more time for healthy cooking and makes smoking less affordable. Furthermore, please only concentrate on studies using similar study populations, rather than studies using data from the general population.

- The cross-sectional design of this study is a huge limitation. The authors should discuss this as a limitation in their discussion. Another limitation that should be added is that, next to the risk factors, employment status is self-reported too.

- Could the authors emphasize on the implications of this study and what are their suggestions for further research?

Reviewer #2: The authors explored the co-occurrence and clustering of risky health behaviors in currently and/or formerly unemployed men and women. This manuscript is based on an impressive dataset and fills a gap of knowledge on clustering of these risky health behaviors in the unemployed, thus making a valuable contribution to the existing literature. However, there are issues that need to be addressed. In my opinion, the most important one is the brief description of the independent variables that quantify the risky health behaviors. The manuscript can be improved substantially when the authors provide more information on the questionnaires, describe the cleaning process, justify the cut-offs and if necessary, repeat the analyses.

Introduction

Many if the examples that the authors provide in the Introduction are based on studies in American populations. As the social programs, social security systems and health care systems of the limited welfare state in the US is significantly different compared to these systems in Western European countries, please focus your examples on the latter.

In addition, please include one or two lines regarding the differences in men and women as these separate analyses are discussed in the Methods, Results and Discussion sections but not in the Introduction.

Page 7 line 155-165

Is ‘at baseline’ the same as ‘at inclusion’? I.e. were all of these measurements done at the same time in one of the 21 medical centers? If so, please consistent terminology throughout the manuscript.

Page 7 line 155-165

Please provide some details on the questionnaires used or refer to a published manuscript on these questionnaires, especially on food frequency and physical (in-)activity. The marker for an unhealthy diet – consuming fruits or vegetables less than three times per day – might be too unspecific. How does this, for instance, relate to the recommendations by WHO? (>400 grams of fruits and vegetables per day).

The authors define physical inactivity as less than 30 minutes of physical activity, defined by walking or cycling, exercising, gardening and housekeeping, over a regular week. Almost 9 out of 10 participants met this requirement (Table 1), which is comforting, but the differences in physical (in-)activity in leisure time might increase after those 30 minutes. The recommendations for adults is +/- 30 minutes of moderate physical activity per day, for at least 5 days a week, including walking, cycling, housework, gardening etc. Please clarify and justify the chosen cut-off of 30min/week.

What is the definition of ‘smoker’ in this study? How was this assessed (how many cigarettes per day/week)? Were consumers of e-cigarettes or cannabis-users removed from the analysis/set to missing/..?

As this paragraph describes the behavior risk factors and therefore the independent variables of the analyses, I would advise the authors to put some more effort in describing these variables or provides references to the questionnaires used. This is very important for validation of the results. If necessary, the analyses should be repeated using more justified cut-offs.

Page 10 line 122-216

Please rephrase this sentence for clarification. If I read correctly, ‘their exposure’ and ‘they’ refers to participants who were unemployed at baseline with or without past experience of unemployment, but to make this more clear please indicate which group is described. In addition, please clarify the last part of the sentence ‘alcohol abuse becoming more frequent than leisure-time physical activity’.

Page 11 line 237

‘…one, two or three factors significantly lower’ There is a word (‘were’?) missing from this sentence. Also, depending on your chosen significance level (which was not stated in the Methods section) a p value of .03 or .08 in the fully adjusted model is or is not significant.

Page 12 line 253

‘In summary, while the unemployed were overexposed to any risk and to combinations of three and four risks…’ The unemployed (assuming the authors are referring to participants having experienced unemployment of any kind, line 251) were over-exposed to no risk, three and to combinations of three and four risks.

Page 12 line 263

Please add a . after ‘factors'.

Page 13 lines 282-283

‘As expected..’ Please include one or two lines regarding the differences in men and women in the introduction. See also my comment on this above.

Page 14 line 298

‘At the time of the analyses’ Please change to ‘at inclusion’ See also comment on Page 7 line 155-165.

Page 14 line 300-302

‘Prevalence rates of exposure to non or only one risk factor were at the higher end while rates of exposure to two, three or four factors were at the lower end of the distributions described in other Western populations.’ On page 11 line 239-243 the authors state that that the exposure to a single factor was almost the most frequent, followed by those to two, none, three and finally four. Please clarify this difference.

Page 15 line 322-327

The reasons for the low fruit and vegetable intake among all participants and low prevalence of leisure-time physical inactivity might lie in the instruments or cut-offs the authors used to create their dichotomous variables. See also my comments for Page 7 line 155-165.

Page 15 line 336-342

This paragraph mentions two times ‘the(se) observations’ (line 336 and 338/339. Please specify to which ones you are referring to as this is very important to follow the authors’ line of reasoning. The finding that most of ‘these observations’ can also be made (albeit to a lesser extent) in employed participants who have been unemployed in the past, might not necessarily support the view that the increased prevalence and co-occurrence of risky behaviors would be a consequence of unemployment as this association can also be spurious. Please comment on this in the discussion. Finally, please replace ‘participate’ (line 342) by ‘contribute’.

Reviewer #3: PLOS ONE

Ms no PONE-D-19-26293

Line 116-118: The distinction between co-occurrence and clustering is basic. This distinction should be explained more clearly and elaborated a little in order to facilitate for the reader.

Line 122: “Western populations”? The study is from the US. Is it valid for France; I suppose the eating patterns are quite different?

Line 128: Unclear what covers over 85 per cent of the population; the CONSTANCES cohort or the general health insurance system?

Line 129: How many persons are included in the cohort? The reference given (44) is written in 2016 and only half of the planned population was included at that time (the intention was 200 000 persons).

Line 133: What does an inclusion rate of 7.3 per cent mean? What is the denominator?

Line 190: No adjustment for inclusion year (seven years of inclusion as I can see)?

Line 192: How was the clusters identified? Cf the different methods that is described in ref. 44 in the manuscript.

Line 211: “they”, who were they? The unemployed (and which of the types of unemployment)?

Line 254-255: This basic summary has to be written so it easily can be understood, the phrasing “did not exceed what could be expected given the prevalence of each risk factor according to unemployment cateogory” is hard to grasp.

Line 296: There is no discussion of how the high drop-out rate (92.7 per cent as I can figure out) may affect the results. Could the low unemployment rate at the inclusion be due to this?

Line 298: “unemployed at the time of the analyses” – was that really measured? Was not unemployment measured at inclusion?

Line 343: not very good references. One is too old (from 1984) and the other is about gender differences in health, not about gender differences in unemployment, so it is a very indirect reference. Refer to a publication with unemployment statistics.

Line 393: The references are not written according to the Vancouver style (which is recommended in the instructions for the authors).

Reviewer #4: • The article investigates the association between unemployment and the co-occurrence of risky health behaviors (clustering). It uses data form the Constances cohort.

• The article hardly presents anything new. I guess we all know that there’s a higher prevalence or occurrence and even co-occurrence of risky behavior in unemployed populations. I sort of miss a good and convincing story here.

• It would for example be interesting to investigate the association between the clustering of risky behavior and some health outcomes. What is the interaction effect between the different kinds of unhealthy behavior? For instance, is the impact equal to the sum of the impact of smoking + the impact of drinking or is there an interaction effect, the one strengthening the impact of the other (by gender and age group for example). I have the feeling there was a lot more potentiality in the data…

• The authors do not present a clear theoretic framework in which the mechanisms are discussed.

• Selection effects are nowhere mentioned, while it is clear in literature that these are an important dimension explaining part of the association.

• The style of writing still could improve considerably.

6. PLOS authors have the option to publish the peer review history of their article (what does this mean?). If published, this will include your full peer review and any attached files.

Reviewer #1: Yes: Gerrie-Cor Herber

Reviewer #2: No

Reviewer #3: No

Reviewer #4: No

---

## [Author Response · Author response to Decision Letter 0]

31 Jan 2020

Please see the attached file "Response to the reviewers"

---

## [Decision Letter · Decision Letter 1]

2 Mar 2020

PONE-D-19-26293R1

Association between unemployment and the co-occurrence and clustering of common risky health behaviors: Findings from the Constances cohort

PLOS ONE

Dear Dr Plessz,

Thank you for submitting your manuscript to PLOS ONE. After careful consideration, we feel that it has merit but does not fully meet PLOS ONE’s publication criteria as it currently stands. Therefore, we invite you to submit a revised version of the manuscript that addresses the points raised during the review process.

We would appreciate receiving your revised manuscript by Apr 16 2020 11:59PM. To enhance the reproducibility of your results, we recommend that if applicable you deposit your laboratory protocols in protocols.io, where a protocol can be assigned its own identifier (DOI) such that it can be cited independently in the future. For instructions see: http://journals.plos.org/plosone/s/submission-guidelines#loc-laboratory-protocols

A **rebuttal letter** that responds to **EACH** point raised by the academic editor and reviewer(s). This letter should be uploaded as separate file and labeled 'Response to Reviewers'.A **marked-up copy** of your manuscript that highlights changes made to the original version. This file should be uploaded as separate file and labeled 'Revised Manuscript with Track Changes'.An **unmarked version** of your revised paper without tracked changes. This file should be uploaded as separate file and labeled 'Manuscript'.

We look forward to receiving your revised manuscript.

Kind regards,

Brecht Devleesschauwer

Academic Editor

PLOS ONE

Additional Editor Comments (if provided):

Both reviewers appreciated the revisions made by the authors, but highlighted some remaining issues, that can be addressed in a final, minor revision round.

Reviewers' comments:

Reviewer's Responses to Questions

**Comments to the Author**

1. If the authors have adequately addressed your comments raised in a previous round of review and you feel that this manuscript is now acceptable for publication, you may indicate that here to bypass the “Comments to the Author” section, enter your conflict of interest statement in the “Confidential to Editor” section, and submit your "Accept" recommendation.

Reviewer #1: All comments have been addressed

Reviewer #2: (No Response)

2. Is the manuscript technically sound, and do the data support the conclusions?

Reviewer #1: Yes

Reviewer #2: Yes

3. Has the statistical analysis been performed appropriately and rigorously? 

Reviewer #1: Yes

Reviewer #2: Yes

4. Have the authors made all data underlying the findings in their manuscript fully available?

Reviewer #1: No

Reviewer #2: Yes

5. Is the manuscript presented in an intelligible fashion and written in standard English?

Reviewer #1: Yes

Reviewer #2: No

6. Review Comments to the Author

Reviewer #1: Although the manuscript is improved, I still have a few issues to consider. I think the authors are getting off easy sometimes and a few of my questions were not sufficiently answered. One example is that stratified analyses are not presented even though statistical significant interactions with three covariates were found. It might be the results are even stronger in low educated people, but estimates and further explanations are not given, which makes it very hard to interpret these results.

Reviewer #2: The authors made significant changes to the manuscript, which improved the quality of the article substantially. However, the language used might need some improving/polishing.

Other comments:

- This manuscript explores the relationship of risky health behaviors with the (un)employment. The added value of this manuscript is the examination of the extent to which common behavioral factors cluster in unemployed individuals. The introduction provides us with many examples of the association between these risk factors and unemployment, but this part of the manuscript could benefit by focusing on the definitions of co-occurrence and clustering of risky health behaviors (in the very beginning) and the implications for social science or health policies (instead of simply demonstrating the association between health factors and (un)employment).

- Line 171-177 “Leisure-time physical activity was determined by a calculated score ranging from 0 (i.e. being very active) to 6 (being not active at all).” According to the explanation of the scoring in the following sentences, I assume a score of 0 is associated with being not active at all and 6 with being very active. However, as in Table 1 ‘No’ is mainly associated with a favorable outcome, it could be that this variable recoded?

- I understand the aim of the authors to explore clustering of risky behaviors, for which dichotomous variables are needed. This required crude (and sometimes arbitrary) cut-offs for the risk factor-variables and those might be of consequence for your conclusions, especially those on the implications of the study.

- Line 236-240 Please rephrase, as this sentence is quite long and I’m not sure which closing paired bracket belongs to which opening paired bracket.

- The authors report that unhealthy behaviors did not cluster more among participants experiencing unemployment than among those who are employed. They suggest that “preventive strategies addressing risky behaviors need not target the unemployed specifically, because they are overexposed but according to similar patterns of co-occurrence.” Line 345-366. Besides the fact that this sentence might need some polishing, I’m not sure whether I understand the main message of this sentence.

- Please provide chapter or page numbers for reference [65].

7. PLOS authors have the option to publish the peer review history of their article (what does this mean?). If published, this will include your full peer review and any attached files.

Reviewer #1: Yes: Gerrie-Cor Herber

Reviewer #2: No

---

## [Editor Report · Decision Letter 2]

13 Apr 2020

Association between unemployment and the co-occurrence and clustering of common risky health behaviors: Findings from the Constances cohort

PONE-D-19-26293R2

Dear Dr. Plessz,

We are pleased to inform you that your manuscript has been judged scientifically suitable for publication and will be formally accepted for publication once it complies with all outstanding technical requirements.

With kind regards,

Brecht Devleesschauwer

Academic Editor

PLOS ONE

Additional Editor Comments (optional):

Thank you for addressing the final comments.
---

## [Editor Report · Acceptance letter]

22 Apr 2020

PONE-D-19-26293R2 

Association between unemployment and the co-occurrence and clustering of common risky health behaviors: Findings from the Constances cohort 

Dear Dr. Plessz:

I am pleased to inform you that your manuscript has been deemed suitable for publication in PLOS ONE. Congratulations! Your manuscript is now with our production department. 

With kind regards,

on behalf of

Prof. Dr. Brecht Devleesschauwer 

Academic Editor

PLOS ONE